# Diet and Stone Disease in 2022

**DOI:** 10.3390/jcm11164740

**Published:** 2022-08-13

**Authors:** Jessica C. Dai, Margaret S. Pearle

**Affiliations:** UT Southwestern Medical Center, 2001 Inwood Road, WCB3, 4th Floor, Dallas, TX 75390-9110, USA

**Keywords:** nutrition, stone risk, nephrolithiasis

## Abstract

Diet plays a central role in the development and prevention of nephrolithiasis. Although pharmacologic treatment may be required for some patients who are resistant to dietary measures alone, dietary modification may be sufficient to modulate stone risk for many patients. While there is no single specialized diet for stone prevention, several dietary principles and recommendations for stone prevention are supported by practice guidelines, including adequate fluid intake, modest calcium intake, low dietary sodium, and limited animal protein. In this review, we summarized the evidence supporting these dietary recommendations and reviewed the current literature regarding specific dietary components and comprehensive diets for stone prevention.

## 1. Introduction

Stone disease affects 1–13% of the adult population worldwide [1]. Without medical intervention, recurrence rates exceed 30% at 10 years, with even higher rates for recurrent stone formers [2,3]. Although pharmacologic therapy is recommended for some patients, lifestyle and dietary management form the cornerstone of stone prevention for most [4,5]. In this review, we summarized the current literature regarding dietary interventions to reduce stone risk in the adult population.

## 2. Fluids and Stone Risk

High fluid consumption is essential for stone formers of all kinds. Clinical practice guidelines recommend a fluid intake that will achieve a urine volume of at least 2–2.5 L daily to reduce the risk of recurrent stone formation [4,5,6]. Several large prospective observational cohort studies validated the beneficial effect of fluid intake on stone risk. Curhan and colleagues studied 91,731 women aged 30–55 years in the Nurses’ Health Study (NHS) I and 96,245 women aged 25–42 years in NHS II. All participants completed semi-annual food frequency questionnaires every 4 years. Compared to patients in the lowest quintile of fluid intake, those in the highest quintile of fluid intake (>2592 and ≥2769 mL/day for NHS I and NHS II, respectively) had a 32–39% lower risk of incident symptomatic stone events after adjusting for confounders [7,8]. In the Health Professionals Follow-up Study (HPFS), Taylor and associates followed 45,619 men aged 40–75 years who completed food frequency questionnaires every 4 years and found a similar trend; men with the highest fluid intake (>2517 mL/day) were found to have a significantly lower risk of incident stones compared to those with the lowest intake (adjusted RR 0.71, 95% CI 0.59–0.85, *p* < 0.001) [9]. A recent analysis of 439,072 incident stone formers enrolled in the United Kingdom BioBank in which food frequency questionnaires were also utilized, demonstrated a reduced risk of recurrent stone events with greater fluid consumption (HR 0.87 per 200 mL fluid, 95% CI 0.85–0.89, *p* < 0.001) at 3 years [10].

The protective effect of generous water intake was demonstrated in a seminal randomized, controlled trial (RCT) by Borghi and colleagues in which 199 idiopathic calcium oxalate stone formers were randomized to either a high water intake (sufficient to produce more than 2 L of urine daily, study group) or to no specific dietary recommendations (control group). Over the 5-year study period, the risk of stone recurrence in the study group was less than half that of the control group (12.1% vs. 27%, respectively, *p* = 0.008). Furthermore, the study group had significantly greater urine volumes and lower relative supersaturations for calcium oxalate (CaOx), brushite (calcium monohydrate phosphate, CaHPO_4_·2H_2_O), and uric acid (UA), which were sustained over the entire period of follow-up. In addition, the mean time to recurrence was significantly longer in the high fluid group compared to the control group (38.7 ± SD 13.2 months vs. 25.1 ± SD 16.4 months, respectively, *p* = 0.016) [11].

However, it is not clear that all beverages have the same beneficial effect on stone formation as water. Indeed, even variations in water composition were suggested to differentially impact stone risk. Drinking water varies widely in calcium and magnesium content depending on geographic location and source [12,13,14]. For example, mineral water typically contains bicarbonate, calcium, magnesium, and other ions, whereas tap water does not. Several epidemiological studies found no significant correlation between water hardness and incident stone risk [12,15,16]. However, small metabolic studies among both stone formers and healthy individuals found increasing water hardness to be associated with increased urinary calcium excretion among healthy individuals and stone formers, with varying reported effects on urinary oxalate excretion [17,18,19,20,21,22]. Consumption of mineral water was also shown to increase urinary citrate compared to water with low minerality [20,21,22,23,24]. However, no prospective controlled studies evaluated the impact of water composition on clinical outcomes.

A number of other beverages were analyzed for their effect on stone formation, and these studies are summarized in the following sections.

### 2.1. Fruit Juice

Fruit juices have received a lot of attention with regard to mitigating stone risk because of the naturally high citrate content of fruits and vegetables. Citrate acts as an inhibitor of calcium oxalate and calcium phosphate stone formation through several mechanisms [25]. First, citrate complexes with calcium in the urine, thereby reducing urinary ionized calcium and decreasing supersaturation of calcium salts. In addition, citrate directly inhibits calcium oxalate and calcium phosphate crystal growth and aggregation. Urinary citrate excretion is determined by systemic acid-base balance, with acidosis reducing citrate production and excretion from the kidney and alkalosis promoting a citraturic response. While fruit juices, in general, have been presumed to constitute a good source of volume, citrate, and alkali, all of which should reduce the risk of stone formation [26], the alkali content of the juice is actually the determining factor in the citraturic response. Orange juice was shown in a small two-phase metabolic study to increase urine pH and citrate compared to water due to its high content of potassium citrate, which provides an alkali load to the renal proximal tubule that enhances citrate production and excretion [27].

The high citrate content of lemons initially made lemonade a promising natural alternative to potassium citrate for hypocitraturic stone formers [28,29]. However, the citrate in lemon juice is primarily in the form of citric acid. As such, when citrate is absorbed from the intestinal tract and converted to bicarbonate, the hydrogen ion generated from citric acid neutralizes the bicarbonate, resulting in no net alkali load. In this case, the citraturic response is limited only to the citrate that escapes conversion to bicarbonate and is excreted directly into the urine in its unmetabolized state.

Despite the lack of physiologic data supporting the benefit of lemonade therapy, the literature on the effect of lemon juice or lemonade therapy on stone risk is conflicting, with some metabolic and retrospective studies demonstrating increased urinary citrate and others demonstrating no changes in urinary citrate or pH with lemonade therapy (Table 1).

Based on short-term metabolic studies, lemonade therapy does not appear to provide an alkali load, as shown by no change in urine pH, and the effect on urinary citrate is mixed. Of note, these studies differ in the populations studied: healthy volunteers, stone formers, or hypocitraturic stone formers. Furthermore, retrospective studies are compromised by lack of dietary data and, in some cases, concomitant use of potassium citrate in patients on lemonade therapy. In the only RCT with stone formation as an outcome, Ruggenenti and associates randomized 203 recurrent CaOx stone formers maintained on a controlled, low animal protein, low-salt, normal-calcium diet to 60 cc of lemon juice twice daily or with no additional intervention. At the conclusion of the 2-year trial, an intention-to-treat analysis showed no significant difference in stone recurrence rates between the two groups (HR 0.62, 95% CI 0.36–1.10, *p* = 0.107). However, this study was limited by the concurrent administration of potassium citrate in hypocitraturic patients as well as poor compliance with lemonade therapy (48% at 2 years) [38]. Consequently, this study is unable to convincingly support or refute the benefit of lemonade therapy.

While physiologically, orange juice (OJ) has been shown in short-term metabolic studies to promote alkaline urine and a citraturic response [27,39], the recommendation for lemonade therapy remains prevalent among practitioners. OJ was directly compared to lemonade therapy in two metabolic studies (Table 1) [31,37]. In a three-phase randomized crossover study, Odvina and colleagues found that among healthy volunteers and stone formers, OJ had a greater citraturic response and raised urine pH more than lemonade when both juices were matched for citrate content. Furthermore, OJ was associated with a lower relative saturation ratio for CaOx and UA compared to lemonade, despite an increase in urinary oxalate with orange juice but not with lemonade [31]. Large and colleagues compared urinary stone risk factors in healthy volunteers who underwent a four-phase study on a self-selected diet consisting of a water phase, a phase with each of 2 low-calorie OJ drinks, and a Crystal Light^®^ lemonade beverage phase. They found no significant favorable changes in urinary stone risk factors in any phase, except for a significant increase in urinary pH with one of the low-calorie OJs, although without a concomitant increase in urinary citrate [37]. On the other hand, pooled analysis of the NHS I, NHS II, and HPFS revealed that consumption of OJ at least once daily was associated with a 12% risk reduction in incident stone formation, compared to less than once weekly intake [40]. Overall, OJ appears to have the most consistent, although not unequivocal, benefit, and lemonade therapy is of doubtful benefit. Definitive recommendations await prospective RCTs in which stone recurrence rate is the primary outcome.

Other fruit juices were also explored with regard to their effect on stone formation. Grapefruit juice was shown in several short-term metabolic studies to promote a number of favorable changes in urinary parameters, including an increase in urinary citrate [39,41,42]. However, one study additionally demonstrated an increase in urinary oxalate such that the net effect was no change in urinary saturation of calcium oxalate [41]. Furthermore, the epidemiologic data from NHS I and HPFS found a significant *increase* in the risk of incident stone formation (OR1.44, 95% CI 1.09–1.92 and OR 1.37, 95% CI 1.01–1.85, respectively) with each daily 8 oz serving of grapefruit juice [43,44].

The role of cranberry juice on stone risk was also investigated in small metabolic studies. Among healthy individuals and CaOx stone formers on a controlled, metabolic diet, daily consumption of cranberry juice significantly decreased urinary pH and increased urinary oxalate and calcium compared to water, thereby increasing urinary saturation of calcium oxalate and stone risk [45,46]. On the other hand, Rodgers and associates found favorable changes in urinary stone risk factors in a two-phase, randomized crossover study among healthy male volunteers on a self-selected diet comparing cranberry juice versus water. They observed an increase in urinary citrate and a decrease in urinary oxalate and phosphate, resulting in a decline in CaOx supersaturation with daily cranberry juice compared to water [47]. Given the inconsistencies in the findings of these studies, cranberry juice cannot be recommended for stone prevention.

Studies of other juices are limited. Metabolic studies in women found that daily consumption of 0.5–1 L of apple juice significantly increased urinary citrate without a significant impact on the supersaturation of CaOx, calcium phosphate, or UA [39]. However, pooled analysis of the NHS I, NHS II, and HPFS did not demonstrate a benefit of apple juice with respect to the risk of incident stone formation [40]. Fresh tomato juice was shown to be a good source of citrate (82.4 mmol/L) but was not associated with a reduction in stone risk in large observational cohort studies [40,48]. Finally, coconut juice, which provides 13.8 mEq/L of alkali, comparable to that of Crystal Lite^®^, yielded increases in urinary citrate, potassium, and chloride compared to tap water when consumed at 2 L daily [37,49]. However, no studies evaluating the clinical impact of these changes have been reported.

Ultimately, the benefit of fruit juice consumption for stone prevention must be carefully balanced with the sugar and caloric content. In potentially therapeutic volumes, sugar content and calories may surpass the amount that would otherwise be consumed as part of a balanced diet. Moreover, the effects of simple sugar consumption on urinary parameters and risk of metabolic syndrome, as well as downstream effects on stone risk, must also be considered.

### 2.2. Tea and Coffee

Stone formers are typically counseled to avoid tea, a common source of dietary oxalate. However, large observational studies worldwide have actually found a potential preventative role for tea consumption. In the United States, NHS I, NHS II, and HPFS found an 8–14% reduction in the risk of incident stone disease with the consumption of at least one 8 oz serving of tea daily [40,43,44]. Among 439,072 patients in the United Kingdom, greater tea consumption was also associated with a decreased risk of incident stones (HR 0.95 per 200 cc/day, 95% CI 0.92–0.99) [10]. Several studies among healthy Chinese individuals demonstrated a similar relationship [50,51,52,53].

Specific tea types should also be considered, as teas vary in oxalate content [54]. A retrospective study of female hypercalciuric stone formers by Rode and colleagues found a lower prevalence of calcium oxalate monohydrate stones among green tea drinkers compared to non-tea drinkers (0 vs. 42%), despite similar urine volume, calcium, oxalate, citrate, and urate between the two groups [55]. It was hypothesized that the antioxidant effects of catechins in green tea, shown in the in vitro and animal studies to reduce urinary oxalate excretion and calcium oxalate stone formation, may account for the protective effect [56]. Seiner and co-workers found that among healthy men, 1.5 L daily of black tea significantly raised urinary citrate without significantly altering any other urinary parameters, compared to a non-oxalate fruit tea control. This effect may be due in part to the relatively low bioavailability of oxalate from tea and further mitigated by simultaneous consumption of milk [56,57].

Coffee consumption was also suggested in epidemiologic studies to have a protective effect against first-time stone formation. In both NHS I and HPFS, greater coffee consumption was associated with a reduced risk of incident stone formation (RR 0.90, 95% CI 0.85–0.95 and RR 0.90, 95% CI 0.85–0.96, respectively, per 8 oz caffeinated coffee); decaffeinated coffee demonstrated a similar effect [43,44]. Littlejohns and associates observed an 8% risk reduction in incident stone formation with each 200 cc serving of coffee/day among individuals in the United Kingdom [10]. Although the protective effect of coffee is thought to be primarily related to the caffeine content and the associated diuretic effect, other bioactive compounds such as *Trigonelline* were also suggested to play a role [56].

### 2.3. Other Beverages

Adequate water intake has proven challenging for many stone formers. However, consumption of more palatable beverages such as sodas, colas, or sports drinks may negatively affect urinary parameters and increase stone risk. The findings of several small metabolic studies evaluating the impact of these beverages are summarized in Table 2.

Epidemiological studies (NHS I, NHS II, and HPFS) demonstrated a positive association between soda consumption and stone risk. Pooled data showed that consumption of at least one sugar-sweetened soda daily was associated with a 23–33% greater risk of stone formation compared to consumption of less than one serving weekly. In contrast, consumption of artificially sweetened colas and non-colas was only marginally and inconsistently associated with increased stone risk [40].

In the only prospective randomized trial evaluating the effect of soft drinks on the risk of recurrent stone formation, Shuster and colleagues found that cessation of soft drink intake among regular soda drinkers (≥160 cc/day) resulted in a 6.4% lower risk of stone recurrence at 3 years. Interestingly, recurrence was significantly lower among those consuming sodas primarily acidified by phosphoric acid (e.g., dark colas) but not for those consuming sodas primarily acidified by citric acid (e.g., clear sodas) at baseline [63]. A subsequent study by Eisner and colleagues assessing the alkali content of diet soft drinks found that diet citrus sodas contained significantly greater alkali than dark colas, which could account for the differential outcome observed by Shuster and colleagues with phosphoric acid based- versus citric acid-based soft drinks [64]. Of note, however, metabolic studies evaluating Fresca^®^ and Diet Sunkist Orange^®^ soda failed to demonstrate significant favorable changes in urinary parameters (Table 2) [60,62].

## 3. Calcium

Historically, dietary calcium restriction was recommended for the management of hypercalciuria and calcium-based stones for obvious reasons. However, more recent studies suggest that calcium restriction may not prevent recurrent kidney stones and could contribute to bone loss. Indeed, according to current guidelines, the recommended daily allowance of calcium of 1000–1200 mg/day holds for stone formers as well [4,5].

Physiologically, intestinal calcium binds oxalate, thereby decreasing intestinal oxalate absorption. A reduced dietary calcium intake results in greater intestinal absorption of uncomplexed oxalate and a subsequent increase in urinary oxalate excretion. Therefore, on a calcium-restricted diet, the lower urinary calcium excretion may be counterbalanced by the increase in urinary oxalate excretion. By using radiolabeled [^13^C_2_]oxalate, increasing daily calcium intake by 200–1200 mg was shown to decrease overall gastrointestinal oxalate absorption from 17% to 2.6%, thereby leading to a decline in mean urinary oxalate excretion [65].

Several large observational cohort studies showed that lower dietary calcium intake is associated with an increased risk of nephrolithiasis [66]. According to the NHS I study, women in the highest quintile of calcium intake had a lower risk of incident stone formation compared to those in the lowest quintile (adjusted RR 0.65, 95% CI 0.50–0.83, *p* = 0.005) [7]. This relationship held true for younger women in NHS II (adjusted RR 0.73, 95% 0.59–0.90, *p* = 0.007) and for men < 60 years old in the HPFS (RR 0.69, 95% CI 0.56–0.87, *p* = 0.01) [8,9]. Of note, however, Heller and colleagues performed a 2-phase metabolic study matching the lowest and highest quintiles of calcium intake from these epidemiologic studies along with implementing modest oxalate restriction and found that oxalate restriction prevented the rise in urinary oxalate associated with the low calcium diet. Furthermore, they showed that the increase in urinary calcium associated with the high calcium diet was associated with no change in urinary saturation of calcium oxalate because of a concomitant increase in the intake of stone-protective factors such as fluids, potassium, and magnesium [67].

A landmark RCT in recurrent calcium oxalate stone-forming hypercalciuric men provided the first Level I evidence against the recommendation for calcium restriction. Borghi and colleagues randomized 120 participants to a “normal calcium” diet (30 mmol/day) with low animal protein (52 g/day) and low sodium (50 mmol/day) intake or a “low calcium” diet (10 mmol/day) with no additional dietary restrictions. At 5 years, they observed a 49% risk reduction in stone recurrence in those on the “normal calcium” diet (95% CI 0.24–0.98, *p* = 0.04) compared to those on the “low calcium” diet. Although urinary calcium declined significantly in both groups, urinary oxalate increased on the low-calcium diet but not on the normal calcium diet [68]. Unfortunately, the independent effects of modified protein and salt intake could not be distinguished from that of dietary calcium intake alone in the “normal calcium” group.

Unlike dietary calcium, calcium supplementation may actually increase the risk of stones in some patients. Among older women in NHS I, supplemental calcium was associated with a significantly increased risk of incident stones (adjusted RR 1.21, 95% CI 1.02–1.41, *p* = 0.03) [7]. Notably, over two-thirds of the women took their calcium supplements outside of mealtimes or at meals with low oxalate content, which limits the beneficial effect of complexing intestinal oxalate, thereby leaving the increase in urinary calcium unopposed [7]. The importance of calcium supplement timing was further underscored in a metabolic study by Domronkitchairporn and co-workers, which showed a decrease in urinary oxalate excretion only when supplements were taken with meals and an increase in CaOx activity product only when calcium supplements were taken apart from meals [69].

The Women’s Health Initiative (WHI) study further demonstrated the risk of calcium and vitamin D supplementation in a cohort of postmenopausal women. In this large RCT, 36,282 participants were randomized to either 1000 mg elemental calcium and 400 IU vitamin D daily in divided doses or to placebo. The incidence of kidney stones was higher among those receiving supplementation (HR 1.17, 95% CI 1.02–1.34, *p*-value not reported), although neither baseline calcium intake nor baseline supplement use was associated with the risk of stones [70]. Thus, general recommendations for stone prevention should include a diet with modest calcium intake (the recommended daily allowance). If calcium and vitamin D supplementation is indicated, these supplements should be consumed *with* meals, if possible. 

## 4. Sodium

Dietary sodium alters the risk of stone formation through its effect on urinary calcium excretion. Greater sodium intake expands extracellular fluid volume and inhibits renal tubular calcium reabsorption, thus increasing urinary calcium [71]. For each 100 mmol (2300 mg) increase in dietary sodium, urinary calcium excretion increases by roughly 1 mmol (40 mg) in healthy adults; among calcium stone-formers, the increase in urinary calcium excretion is even greater [72].

Small metabolic studies showed that a high sodium diet (250 mmol/day) increases urinary sodium, calcium, and pH and decreases urinary citrate, compared to a low-sodium diet [71]. A randomized trial of 210 hypercalciuric stone formers who consumed a control diet or a low-sodium diet with modest calcium intake found that the low-sodium diet was associated with a 70% reduction in urinary sodium and chloride from baseline. Notably, urinary calcium and oxalate excretion decreased significantly with sodium restriction, and the proportion of patients rendered normocalciuric was nearly double that of the control diet (62% vs. 34%) [73].

The clinical significance of dietary sodium was borne out by large observational cohort studies of older women. In NHS I, sodium consumption in the highest quintile of intake was associated with a 30% greater risk of incident stone formation than in the lowest quintile (95% CI 1.05–1.62, *p* < 0.001) [7]. Among 78,293 women in the WHI study, the relative risk of incident stones for those with the highest quintile of sodium consumption was 1.61 (95% CI 1.32–1.96, *p* ≤ 0.001), compared to the lowest [74]. Furthermore, a randomized trial by Borghi and colleagues demonstrated a nearly 50% reduction in stone recurrence rate with a low-sodium diet in conjunction with low animal protein and moderate calcium intake compared to a low-calcium diet [68]. Consequently, practice guidelines recommend limiting salt intake for stone formers. The EAU [5] recommends a total daily intake of <3–5 g, while the AUA [4] supports a more stringent daily limit of <2300 g (≤100 mEq).

## 5. Oxalate

Hyperoxaluria is a risk factor for calcium oxalate stone formation by increasing urinary saturation of CaOx, which was shown to predict stone formation [75]. Whereas CaOx supersaturation plateaus above urinary calcium concentrations of 5 mmol/L, there is an exponential rise in CaOx supersaturation with increasing urinary oxalate concentrations up to 0.7 mmol/L [76]. Thus, oxalate intake is a significant dietary target for stone prevention.

Oxalate is both endogenously produced by the liver and consumed through diet. Endogenous oxalate production is further modulated by hydroxyproline (found in collagen-containing meat products and gelatin-containing foods) and ascorbic acid (vitamin C), which may also be influenced by dietary intake [77,78,79]. A controlled, metabolic study evaluating the effect of 2 g of vitamin C daily in normal subjects and stone formers revealed a 20% and 33% increase, respectively, in urinary oxalate with vitamin C compared to placebo [79]. Among men in the HFPS, excess vitamin C consumption (≥1000 mg/day) was associated with a 41% increase in the risk of incident stones (95% CI 1.11–1.80, *p* = 0.1) [9]. In another study of Swedish men, vitamin C supplementation was further associated with a nearly two-fold increased risk of incident stone disease over 11 years of follow-up (multivariate RR 1.92, 95% CI 1.33–2.77); however, the actual dosages consumed were not reported in this study [80].

Typical dietary oxalate intake in the Western diet ranges from 101 to 152 mg/day and is primarily derived from plant-based foods [81,82]. Typical oxalate-rich foods include nuts, chocolate, spinach, potatoes, beets, and avocados. About 5–15% of ingested oxalate is absorbed from the intestinal tract, and this may be affected by oxalate-degrading organisms in the fecal microbiome or by intestinal bypass surgeries that cause fat malabsorption, saponification of intraluminal calcium, and increased intestinal oxalate absorption [83,84,85]. Furthermore, the bioavailability of oxalate varies widely in different foods. Unfortunately, extensive studies assessing bioavailability in different foods are lacking, and oxalate lists are based on the actual oxalate content of foods, without regard to bioavailability. 

Although dietary oxalate is directly correlated with urinary oxalate excretion [86,87], intestinal oxalate absorption depends not only on dietary oxalate intake but also on dietary calcium intake. Diets with higher calcium lead to reduced intestinal oxalate absorption, while restricted calcium diets are associated with enhanced oxalate absorption and increased urinary oxalate excretion [65,88]. Indeed, Holmes and colleagues showed that dietary oxalate intake contributes up to 24–41.5% of urinary oxalate excretion, but on a high-oxalate, calcium-restricted diet, dietary oxalate intake can contribute up to 52.6% of urinary oxalate [84]. The timing of dietary calcium and oxalate intake plays an important role in determining the effect of dietary oxalate on stone risk. In idiopathic calcium oxalate stone formers, increasing dietary calcium at meal and snack times was found to significantly reduce urinary oxalate and CaOx supersaturations, albeit along with increased urinary calcium [89]. This finding was validated in a two-phase, randomized, crossover metabolic study among 32 healthy men who were given either 3 gm of calcium carbonate daily (all at bedtime away from a meal) or 1 gm of calcium carbonate *with* each meal [69]. Although urinary calcium increased in both groups compared to baseline, urinary oxalate significantly *decreased* when calcium carbonate was given with meals but remained unchanged when given at bedtime. Indeed, urinary saturation of CaOx was unchanged when calcium was given with meals but increased when given at bedtime. These findings suggest that taking calcium with meals reduced urinary oxalate and also protected against the increase in urinary calcium seen with calcium supplementation.

Analysis of NHS I, NHS II, and HPFS revealed that among men in the HPFS and older women in NHS I, there was a 21–22% increase in the risk of incident stones with the highest quintile of oxalate intake (median 287–328 mg/day). The greatest risk was found among men with low dietary calcium intake (RR 1.46, 95% CI 1.11–1.93, *p* = 0.008). However, no significant association between oxalate intake and stone risk was found among younger women in NHS II [90]. However, the inability to assess the timing of concomitant calcium intake or calcium supplementation is a significant limitation. 

Of note, no published study to date has demonstrated a significant benefit of dietary oxalate restriction on actual stone recurrence. However, co-ingestion of calcium-containing foods with oxalate-rich meals or snacks can prevent the rise in urinary oxalate and urinary calcium oxalate saturation but with an increase in urinary calcium.

## 6. Protein

Practice guidelines recommend limiting non-dairy animal protein for calcium, uric acid, and cystine stone formers [4]. However, modern Western diets tend to contain an excess of animal protein, which is thought to increase stone risk. Several mechanisms for this were proposed. First, animal protein provides an acid load due to the high content of sulfur-containing amino acids, which decreases urine pH and citrate and increases urinary calcium excretion through bone resorption and/or reduced renal calcium reabsorption [91]. Interestingly, a recent study suggested that the hypercalciuric effect of animal protein is independent of the associated increase in net acid load [92]. Animal protein also provides a purine load that increases urinary uric acid excretion. Uric acid reduces the effectiveness of natural macromolecular inhibitors of calcium oxalate crystallization, thereby reducing urinary inhibitory activity [93].

Metabolic studies demonstrated the potentially lithogenic effect of excess animal protein on urinary parameters. Among healthy individuals, animal protein promotes a significant decrease in urinary pH, citrate, and oxalate, as well as an increase in urinary calcium, UA, sulfate, and phosphorus, compared to equivalent amounts of vegetarian protein or vegetarian and egg protein. Additionally, there is a corresponding increase in poorly soluble, undissociated urinary UA [94]. Among stone formers in controlled metabolic studies, high protein diets also decreased urinary pH and citrate and increased urinary calcium [95,96]. Pooled analyses of the NHS I, NHS II, and HPFS studies further corroborate these findings [97]. Specific protein types may also have unique effects: fish protein, with its higher purine content, is associated with greater urinary UA excretion than beef or chicken, although beef appears to be associated with the highest saturation index for CaOx, suggesting that purine content alone does not account for the lithogenic effect of animal protein [98].

Despite the findings of metabolic and epidemiologic studies, there remains a lack of definitive clinical data to support animal protein restriction for stone prevention. Among women in NHS I and NHS II, there was no association between animal protein intake and stone risk [7,8]. In the HPFS, a modest association (adjusted RR 1.38; 95% CI 1.05–1.81, *p* = 0.03) was found only among men with body mass index < 25 kg/m^2^ [9]. Ferraro and colleagues evaluated dairy, non-dairy animal, and vegetable protein intake in a pooled analysis of NHS I, NHS II, and HFPS participants and found a small but non-significant increase in stone risk among individuals in the highest quintile of non-dairy animal protein intake (HR 1.10, 95% CI, 1.00 to 1.21, *p* = 0.20). There was no association between vegetable protein consumption and stone risk. Notably, among younger women in NHS II, greater consumption of dairy protein was actually associated with a *decreased* risk of incident stones (HR 0.84, 95% CI 0.73–0.96, *p* < 0.01) [97].

Contradictory findings were reported by RCTs. Borghi and associates reported a 49% reduction in risk of stone recurrence among calcium oxalate stone formers on a low-animal-protein, low-sodium, and normal-calcium diet, compared to those on a calcium-restricted diet, but the independent effect of animal protein on stone recurrence rates was not evaluated [68]. Hiatt and colleagues randomized 99 calcium oxalate stone formers to a low animal protein (56–64 g/day), high-fiber diet, or the usual diet. The low-protein diet and high-fiber diet were associated with a *higher* risk of recurrent stones (RR 5.6, 95% CI 1.2–26.1, *p*-value not reported) compared to the control group. However, 21% of patients were lost to follow-up by the end of the 4-year study period, limiting the validity of the study [99]. Moreover, neither study evaluated animal protein intake alone. A prospective randomized trial by Dussol and colleagues of idiopathic calcium stone formers on a low-animal-protein diet (<13% total energy from protein), a high fiber diet (>25 g/day), or a control diet found that neither a low-protein diet nor a high fiber diet decreased the risk of stone recurrence relative to the control group at 4-year follow-up; however, this trial was also limited by a high drop-out rate (46%) and must be interpreted with caution [100].

## 7. Sugars and Carbohydrates

The published literature on carbohydrates and urinary stone risk has been limited to the evaluation of simple carbohydrates. Several studies demonstrated that ingestion of simple sugars such as glucose or sucrose promotes a hypercalciuric response among healthy individuals and stone formers, which appears to be independent of insulin release [101,102,103,104]. A similar hypercalciuric effect occurs with sugar substitutes such as xylitol [103]. Among healthy individuals, some studies found that daily ingestion of fructose increases urinary oxalate excretion and lowers urinary pH and magnesium [105], while others demonstrated no changes in 24 h urinary oxalate, calcium, or uric acid excretion [106].

The impact of sugar consumption on stone risk was studied in large prospective cohort studies. Among women in the NHS I and NHS II, greater dietary sucrose intake was associated with a higher risk of incident stones (adjusted RR 1.52, 95% CI 1.18–1.96, *p* = 0.001, and RR 1.31, 95% CI 1.07–1.60, *p* = 0.1, respectively) [7,8]. Among patients in NHS I, NHS II, and HPFS, greater fructose intake was independently associated with a 27–37% increased risk of incident kidney stones. Notably, non-fructose carbohydrates were not associated with increased stone risk [107]. Given the increasing prevalence of fructose-sweetened foods in the Western diet, these effects may have a significant clinical impact and warrant further study.

## 8. Fats

Polyunsaturated fatty acids (PUFA) are believed to affect stone formation through their effects on urinary calcium and oxalate. n-6 PUFAs such as arachidonic acid (AA) are present in higher quantities in cell membrane phospholipids of idiopathic calcium stone formers, and these were shown to increase prostaglandin E_2_, which modulates urinary calcium handling resulting in hypercalciuria [108,109]. In contrast, n-3 PUFA (“omega 3 fatty acids”) such as eicosapentaenoic acid (EPA) and docosahexanoic acid (DHA), are thought to reduce stone risk by being competitively incorporated into cell membrane phospholipids in place of AA, thereby reducing urinary calcium and oxalate excretion [110,111,112].

Metabolic studies of PUFA have all been relatively small (*n* = 12–88) and vary in the dose, frequency, and duration of use, limiting meta-analyses. Published findings regarding the effect of n-3 PUFA on urinary parameters are conflicting [109]. However, several studies demonstrated that EPA and DHA supplementation decreases urinary calcium excretion among stone formers [110,111,113,114,115]. Few studies evaluated the effect of n-6 PUFA on urinary parameters [109].

None of the literature definitively supports the clinical benefit of PUFA. Yasui and co-workers administered 1.8 g/day of purified EPA supplement to 29 calcium oxalate stone formers and found a significantly lower rate of recurrent stone events during 3 years of supplement use (0.07 stone events/year) compared to before the initiation and after discontinuation of supplementation (0.23 and 0.17 stone events/year, respectively) [116]. In NHS I, NHS II, and HPFS, Taylor and colleagues found no association between AA or linoleic acid (an AA precursor) intake and risk of incident stones among the three cohorts. Among the older female cohort (NHS I), greater EPA and DHA intake were actually associated with a greater risk of incident stones (adjusted RR 1.28, 95% CI 1.04–1.56, *p* = 0.04), but this trend was not observed for younger women (NHS II) or men (HPFS). Although the study also found no significant association between fish oil supplementation and incident stone risk, the study was underpowered for this endpoint [108]. To date, no RCTs have evaluated the effect of PUFA on stone events.

## 9. Specific Diet Types

Many stone patients already follow specific diets as part of a weight loss or health program. Understanding the effects of popular diets on stone risk is therefore essential. Low carbohydrate, high-protein diets (e.g., Atkins, Zone diet, ketogenic diet); the Dietary Approaches to Stop Hypertension (DASH) diet; and the Mediterranean diet are among the most well-studied diets for stone formers.

Low-carbohydrate, high protein diets are popular for weight loss but have an unfavorable effect on urinary parameters. Studies by Reddy and associates and Friedman and colleagues showed higher urinary calcium in patients on low-carbohydrate, high-protein Atkins-type diets, along with higher urine volume, lower urinary citrate, and lower urinary pH (Table 3) [117,118]. However, in the latter study, the diet was self-directed by participants; therefore, the degree of carbohydrate restriction and protein intake is not known.

The DASH diet was designed to control hypertension, and it is high in fruits, vegetables, nuts, whole grains, and low-fat dairy proteins and low in saturated fat, cholesterol, refined grains, sugar, and meats. Several studies evaluated the effect of the DASH diet on urinary parameters (Table 3). Although the DASH diet increases urinary oxalate, it does not increase CaOx or UA supersaturation, likely due to an associated increase in urine pH and urinary inhibitors such as citrate and magnesium [119,120,121]. Among patients in NHS I, NHS II, and the HPFS, a diet most closely resembling the DASH diet was independently associated with a significantly lower risk of incident stones (adjusted RR 0.58, 0.60, and 0.55, respectively) [122].

A heart-healthy diet, the Mediterranean diet, has also been investigated for its impact on kidney stone risk. The Mediterranean diet is a primarily plant-based diet high in unrefined grains, legumes, vegetables, fruit, and olive oil, with modest dairy and fish consumption and limited meat consumption. Leone and colleagues studied 16,094 young and middle-aged Spanish individuals who completed longitudinal dietary questionnaires. After a mean follow-up of 9.6 years, there was a 36% reduced risk of incident stones among those with the greatest adherence to a Mediterranean diet (HR 0.64, 95% CI 0.48–87, *p* = 0.01) [123]. The benefit of this diet has been postulated to be related to a lower risk of CaOx crystallization risk on this diet [124]. In contrast, Soldati and co-workers found no association between adherence to a Mediterranean diet and stone events in 478 obese Caucasian subjects [125]. Thus, the benefit of a Mediterranean diet remains unclear.

**Table 3 jcm-11-04740-t003:** Summary of studies evaluating the effect of popular diets on urinary parameters.

Study	Study Type	Study Length	Subjects	Diet Type	Findings
Reddy et al., 2002 [117]	2-phase metabolic study	1 week usual diet + 2 week induction phase + 6 week maintenance phase	10 healthy volunteers	Atkins-type low-carbohydrate, high-protein diet (self-directed) vs. usual diet control Induction phase: severe carbohydrate restriction Maintenance phase: moderate carbohydrate restriction 3L fluid intake for all phases	Induction phase: Decreased urinary pH (by 0.53, *p* < 0.01) Increased undissociated urinary uric acid saturation (>2x, *p* < 0.01) Increased net urinary acid excretion (by 56 mEq/day, *p* < 0.001) Decreased urinary citrate (by 314 mg/day, *p* < 0.01) Increased urinary calcium (by 98 mg/day, *p* < 0.001) Maintenance phase: Decreased urinary pH (by 0.52, *p* < 0.001) Increased undissociated urinary uric acid saturation (>2x, *p* < 0.01) Increased net urinary acid excretion (by 51 mEq/day, *p* < 0.001) Decreased urinary citrate (by 182 mg/day, *p* < 0.05) Increased urinary calcium (by 88 mg/day, *p* < 0.01)
Friedman et al., 2012 [118]	Secondary analysis of a multi-center, RCT	24 months	307 obese patients	Low-carbohydrate, high-protein diet vs. low-fat diet	Increased urine volume at 12 months (relative increase 438 cc, 95% CI 181–696 cc, *p* < 0.01) and 24 months (relative increase 268 cc, 95% CI 1–535 cc, *p* < 0.05) on low-carbohydrate, high-protein diet Increased urinary calcium at 3 months (relative increase 36.1%, 95% CI 15–61.1%, *p* < 0.001) and 12 months (relative increase 35.7%, 95% CI 10.7–66.2%, *p* < 0.01) on low-carbohydrate, high-protein diet
Taylor et al., 2010 [119]	Cross-sectional study	Not reported	3426 participants in the Health Professionals Follow-up Study, Nurses’ Health Study I, and Nurses’ Health Study II	DASH diet, as measured by a 7-component DASH score, assigned from food frequency questionnaires to indicate DASH adherence	For Health Professionals Follow-up Study * Increased urinary oxalate (*p* < 0.001), citrate (*p* = 0.004), uric acid (*p* = 0.04), sulfate (*p* < 0.001), potassium (*p* < 0.001), phosphate (*p* = 0.002), pH (*p* = 0.01), volume (*p* < 0.001) Decreased supersaturation of uric acid (*p* = 0.02) For Nurses’ Health Study I * Increased urinary calcium (*p* = 0.05), oxalate (*p* = 0.03), citrate (*p* = 0.002), sulfate (*p* = 0.01), potassium (*p* < 0.001), magnesium (*p* < 0.001), pH (*p* < 0.001), volume (*p* < 0.001) Decreased supersaturation of calcium oxalate (*p* < 0.001) and uric acid (*p* < 0.001) For Nurses’ Health Study II * Increased urinary oxalate (*p* = 0.001), citrate (*p* = 0.008), sulfate (*p* < 0.001), potassium (*p* < 0.001), magnesium (*p* < 0.001), pH (*p* < 0.001), volume (*p* < 0.001) Decreased supersaturation of calcium oxalate (*p* = 0.003) and uric acid (*p* < 0.001)
Noori et al., 2014 [120]	RCT	8 weeks	57 recurrent stone formers with hyperoxaluria—41 completed trial	DASH diet vs. low-oxalate diet	As-treated analysis: Non-significant trend toward increased urinary oxalate excretion in DASH diet (by 9 mg/day, *p* = 0.08) Trend toward decreased calcium oxalate supersaturation in DASH group (by 1.24, *p* = 0.08) Non-significant increase in urinary citrate, magnesium, and pH in DASH group
Maddahi et al., 2020 [121]	Cross-sectional study	Not reported	265 men with nephrolithiasis	DASH diet pattern, as measured by food and nutrient-based DASH scores assigned from food-frequency questionnaires to indicate DASH adherence	Highest tertile of food-based DASH diet score: Lower odds of hypocitraturia (OR 0.17, 95% CI 0.06–0.35, *p* < 0.001) Lower odds of hypercalciuria (OR 0.25, 95% CI 0.11–0.54, *p* < 0.001) Lower odds of hypercreatininuria (OR 0.38, 95% CI 0.17–0.82, *p* = 0.01) Highest tertile of nutrient-based DASH diet adherence: Lower odds of hypocitraturia (OR 0.07, 95% CI 0.02–0.22, *p* < 0.001) Lower odds of hypercalciuria (OR 0.17, 95% CI 0.07–0.41, *p* < 0.001) Lower odds of hyperuricosuria (OR 0.42, 95% CI 0.18–0.95, *p* = 0.04) Lower odds of hypercreatininuria (OR 0.34, 95% CI 0.14–0.83, *p* = 0.01)
Prieto et al., 2019 [124]	Cross-sectional study	Not reported	267 Spanish patients	Mediterranean diet, as measured by MDS to indicate diet adherence	Decreased risk of calcium oxalate crystallization (PR 0.51, 95% CI 0.26–0.87, *p* = 0.012) with greater adherence to Mediterranean diet Non-significant decrease in risk of uric acid crystallization (PR 0.77, 95% CI 1.12–l.46, *p* = 0.069) with greater adherence to Mediterranean diet

RCT = randomized controlled trial; mEq = milliequivalents; DASH = Dietary Approaches to Stop Hypertension; MDS = Mediterranean Diet Score; OR = odds ratio; PR = prevalence ratio; CI = confidence interval, * All comparisons of highest quintile vs. lowest quintile.

## 10. Conclusions

Dietary modification is an essential component of any stone prevention program, and dietary management is appropriate for all stone formers regardless of added pharmacologic therapy. However, the strength of evidence regarding general dietary recommendations is highly variable. The highest-level evidence supports high fluid intake, particularly water, as well as the combined implementation of modest dietary calcium intake with limited animal protein and sodium. The role of oxalate restriction and isolated animal protein restriction remains unclear. However, until future studies provide further high-level evidence to support specific dietary measures, the combined findings of metabolic studies and prospective cohort studies will have to suffice to guide our recommendations.

## Figures and Tables

**Table 1 jcm-11-04740-t001:** Summary of studies on the impact of lemon juice and lemonade on urinary parameters.

Study	Study Type	Study Length	Subjects	Intervention	Diet	Findings
Seltzer et al., 1996 [30]	Single-arm, metabolic study	6 days	12 hypocitraturic calcium stone formers	8 glasses of lemonade daily (4 oz lemon juice in 2 L tap water + sweetener)	Counseled to maintain daily urine output ≥2 L, sodium and protein-restricted diet	Increase in mean urinary citrate (by 204 mg/day, *p* < 0.001) 7/12 patients rendered normocitraturic
Odvina, 2006 [31]	3-phase, randomized, crossover metabolic study	1 week/phase + 3-week washout between phases	9 healthy volunteers + 4 stone formers	(1) Lemonade—400 cc reconstituted frozen concentrate, 3×/day (2) OJ—400 cc reconstituted frozen concentrate, 3×/day (3) Distilled water—400 cc, 3x/day	Controlled metabolic diet: 400 mg calcium 150–200 mg oxalate 800 mg phosphorus 100 mEq sodium 50 mEq potassium 200 mg magnesium Total fluid 3 L	No change from water phase in urinary citrate or pH with lemonade consumption Increased urinary pH (by 0.6, *p* < 0.05), citrate (by 88 ± 30 mg/240 cc *p* < 0.05), magnesium (by 25 mg/240 cc, *p* < 0.0001), potassium (by 47 mg/240 cc, *p* < 0.0001), and oxalate (by 4 mg/240 cc, *p* < 0.005) with OJ Lower net acid excretion (by 38.2 mEq/240 cc, *p* < 0.0001) undissociated UA (by 107 mg/240 cc, *p* < 0.0001), and relative supersaturation of brushite (by 0.5/240 cc, *p* < 0.001) with OJ
Penniston et al., 2007 [32]	Retrospective cohort study	9 years (1996–2005)	100 calcium oxalate stone formers 37 concurrently taking KCit (20–90 mEq/day)	4 oz lemon juice in water or 32 oz low-sugar/sugar-free lemonade daily	Self-selected	Increased urinary citrate among lemonade group (by 203 ± 45 mg/day, *p* < 0.001) and lemonade + KCit group (by 346 ± 45 mg/day, *p* < 0.001) Increase in urinary pH among lemonade group (by 0.24 ± 009, *p* = 0.006) and lemonade + Kcit group (by 0.55 ± 0.10, *p* < 0.001)
Kang et al., 2007 [33]	Retrospective cohort study	8 years (1995–2003)	11 stone formers with hypocitraturia + 11 age/sex-matched stone formers on Kcit (40 mEq/day)	120 cc concentrate lemon juice in 2 L water daily	Self-selected	Increase in urinary citrate in lemonade group (mean increase 383 mg/day, *p* < 0.05) and Kcit group (mean increase 482 mg/day, *p* < 0.0001)). Increase in urinary citrate was greater in Kcit group than lemonade group (*p* < 0.05) Significant increase in urinary pH (by 0.6, *p* < 0.05) and uric acid (by 171 mg, *p* < 0.05) in Kcit group only
Koff et al., 2007 [34]	2-phase, randomized crossover metabolic study	3 days controlled diet + 5 days/phase + 2 week washout between phases	21 stone formers	(1) 3 servings of lemonade daily (30 cc ReaLemon lemon juice + ¾ C water and sweetener) (2) KCit 60 mEq/day	Counseled on low-sodium (2 g/day) and low-protein (65 g/day) diet, 2 L fluid intake	Increase in urinary citrate (by 107 mg/day, *p* = 0.0015) and pH (by 0.38, *p* = 0.0001) and decrease in urine volume (by 0.24 L, *p* = 0.016) from baseline only in Kcit group
Aras et al., 2008 [35]	3-arm, randomized metabolic study	3 months	30 hypocitraturic calcium stone formers	(1) 85 cc of fresh lemon juice in 1 L water daily (2) KCit 60 mEq/day (3) dietary recommendations (3 L/day water, 1200 mg/day calcium, 5 g/day sodium, 1 g/kg/day protein)	Self-selected	Increased urinary citrate (*p* = 0.003 for lemon group, *p* = 0.001 for Kcit and diet groups) and urine volume (*p* = 0.032 for lemon group *p* = 0.035 for Kcit group, *p* = 0.047 for diet group), in all 3 groups from baseline Increased urine pH only in KCit group (5.9 to 6.5, *p* = 0.04)
Cheng et al., 2019 [36]	2-phase, randomized crossover metabolic study	5-day baseline + 5 days/phase + 1 week washout between phases	12 healthy individuals	(1) 2 L water intake (baseline) (2) 2 L regular lemonade daily (3) 2 L diet lemonade daily	Controlled low-oxalate diet 100–150 mEq sodium 800–1200 mg calcium 1–1.2 g protein/kg body weight	Increase in urine volume with both lemonades (*p* = 0.006 for regular lemonade, *p* = 0.002 for diet lemonade) Increased urinary citrate with diet lemonade from baseline (by 104 mg *p* = 0.041) Decreased urinary citrate with regular lemonade from baseline (by 76 mg, *p* = 0.041) No change in urinary pH with either lemonade
Large et al., 2020 [37]	4-phase, randomized crossover metabolic study	7 days/phase + 1 week washout between phases	10 healthy volunteers	(1) 1 L Tropicana 50^®^ OJ + 1 L water/day (2) 1 L Kroger^®^ low-calorie OJ + 1 L water/day (3) 1 L Crystal Lite^®^ lemonade beverage + 1 L water/day (4) 2 L water/day	self-selected	Compared to water: Non-significant increase in urinary pH (by 0.25, *p* > 0.05), citrate (by 155 mg/day, *p* ≥ 0.05), and volume (by 200 cc/day, *p* > 0.05) in Crystal Lite^®^ group Significant increase in urinary pH with Kroger^®^ low-calorie OJ (by 0.74, *p* < 0.05) Non-significant increase in pH with Tropicana 50 ^®^ (by 0.25, *p* > 0.05) Non-significant increase in urinary citrate (by mean of 117–178 mg/day, *p* > 0.05) and potassium (by 24 and 15 mmol/day, *p* > 0.05) in both OJ groups

Kcit = potassium citrate, OJ = orange juice.

**Table 2 jcm-11-04740-t002:** Summary of studies evaluating the effect of sodas, colas, and sports drinks on urinary parameters.

Study	Study Type	Study Length	Subjects	Intervention	Diet	24 h Urine Results
Weiss et al., 1992 [58]	Single-phase metabolic study	48 h	4 healthy volunteers *	3 quarts cola/48 h	Self-selected	Increased urinary magnesium (mean 2.6 mg/day, *p* = 0.048) and oxalate (mean 8.3 mg/day, *p* = 0.003) from baseline Decreased 24 urinary citrate (mean 122 mg/day, *p* = 0.015) from baseline
Rodgers, 1999 [59]	Single-phase metabolic study	24 h	45 healthy volunteers	2 L carbonated cola beverage	Self-selected	Increased urinary oxalate (*p* = 0.05), Tiselius risk index (*p* = 0.03), modified activity product (*p* = 0.03), volume (*p* = 0.03), and relative Brushite supersaturation (*p* = 0.02) among males after cola ingestion Increased urinary oxalate (*p* = 0.002) and phosphate excretion (*p* = 0.01); increased relative uric acid supersaturation (*p* = 0.003); decreased urinary potassium (*p* = 0.009), sodium (*p* = 0.003), and pH (*p* = 0.0001) among females after cola ingestion Greater number and size of COD and COT crystals on scanning electron microscopes after cola ingestion
Passman et al., 2009 [60]	3-phase, crossover metabolic study	5 days/phase + 2 day washout between phases	6 healthy volunteers	(1) Le Bleu^®^ bottled water (2) Fresca (3) Caffeine-free Diet Coke (4) Self-selected diet (baseline control) Liquid volumes based on lean body weight	Controlled metabolic diet (per 2500 kcal) 1000 mg calcium 3.5 g sodium 150 mg oxalate	Increase in urine volume with all 3 beverages vs. control (*p* = 0.0009) Lower urinary uric acid with Fresca consumption (by 125 mg/day, *p* = 0.006) vs. control Decrease in calcium oxalate supersaturation with all 3 beverages vs. control (*p* < 0.05) No difference in any other urinary parameters
Goodman et al., 2009 [61]	2-phase short-term metabolic study	3 days control + 1 week washout period +3 days for assigned sports drink	16 healthy volunteers	(1) 1 quart Performance^®^ sports drink daily (2) 1 quart Gatorade^®^ sports drink daily (3) 1 quart tap water daily (control)	Self-selected	Increase in urinary citrate (by 170.4 mg, *p* = 0.003) and pH (by 0.31, *p* < 0.05) with Performance^®^ only Non-significant increase in citrate (by 84 mg) and urinary pH (by 0.19) with Gatorade^®^
Sumorok et al., 2012 [62]	2-phase, randomized crossover metabolic study	3 days/phase	9 healthy volunteers	(1) 36 oz of water (control) (2) 36 oz (3–12 oz cans) of Diet Sunkist Orange Soda	Patients replicated a self-selected diet from one arm to the other	Significant increase in the supersaturation of calcium phosphate with Diet Sunkist Orange consumption compared to water (*p* = 0.04) No significant changes in urinary pH, citrate, calcium, sodium, oxalate, potassium, or phosphate with Diet Sunkist Orange consumption compared to water

* 1 participant only completed 2 quarts of soda consumption, COD = calcium oxalate dihydrate, COT = calcium oxalate trihydrate.

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
