# Peer review of "Diet and Stone Disease in 2022"

_jcm, 2022, doi:10.3390/jcm11164740_

Round 1
Reviewer 1 Report
This was a very enjoyable paper to read. The paper is logical, and very comprehensive.
I would take care just to ensure that readers can understand the meaning behind certain terms, e.g., brushite [form of calcium phosphate stone].
The only other point I would have is that this entire study appears to be adult-based. There are no references to children, or any studies (however limited) therein. As you will know nearly 30% paediatric stones are metabolic, and it may therefore be worth pointing out to the readers that this is a review of diet and adult stone disease, and that whilst the general recommendations could be translated to a paediatric population, they are not designed to do so.
Great paper!
Reviewer 2 Report
This a a highly comprehensive review on diet and kidney stone disease. Authors base their summary mainly on findings from large epidemiologic trials, such as NHS I, NHS II and HPFS. In addition, they quote RCTs and a few "mechanistic" studies.
This reviewer does not agree with the statement that "the effect of dietary oxalate on actual stone risk remains modest". Nowadays, when most calcium stones are calcium oxalate monohydrate, elevated oxaluria plays a prominent role in increasing CaOx supersaturation which finally predicts stone formation (Prohaska M et al., J Urol 199: 1262-1266, 2018). It has convincingly been demonstrated many years ago that CaOx supersaturation in human urines exponentially rises with increasing urinary Ox concentrations up to a concentration of 0.7 mmol/l, whereas CaOx supersaturation plateaus above urinary Ca concentrations of 5 mmol/l (Roberston WG et al, J Cryst Growth 53: 182-194, 1981). The excretion of oxalate, however, greatly depends on the amount of absorbed Ox which is influenced by the ingestion of calcium simultaneously with Ox-containing products. This was demonstrated by Von Unruh et al. (Ref. 56) as well as - under rather extreme conditions of Ox intake from natural foods - Hess et al. (Nephrol Dial Transplant 13: 2241-2247, 1998) in healthy volunteers. More recently, Sromicki & Hess (Urolithiasis 48: 425-433, 2020) have demonstrated that increasing the intake of calcium from natural foods/fluids simultaneously with any snacks/meals significantly reduced urinary oxalate and CaOx supersaturation in idiopathic calcium stone formers, although urinary calcium increased too. Similar findings highlighting the importance of timing of the intake of calcium (as supplements) were obtained by Domronkitchaiporn et al. (Ref. 60). These findings should be added to the review.
Excess vitamin C consumption and risk of kidney stones: the Swedish study of Thomas et al. (JAMA Intern Med 173: 386-388, 2013) should also be mentioned.
Final paragraph: it should be mentioned that so far NO study ever has demonstrated a significant effect of oxalate restriction on stone recurrences, as oxalate is almost unavoidable. The clinical key point is to reduce gastrointestinal Ox absorption by simultaneous intake of calcium with meals and snacks, most likely together with the reduction of fructose-containing products in order to reduce urinary oxalate form metabolic origin.
Author Response
This a highly comprehensive review on diet and kidney stone disease. Authors base their summary mainly on findings from large epidemiologic trials, such as NHS I, NHS II and HPFS. In addition, they quote RCTs and a few "mechanistic" studies.
This reviewer does not agree with the statement that "the effect of dietary oxalate on actual stone risk remains modest". Nowadays, when most calcium stones are calcium oxalate monohydrate, elevated oxaluria plays a prominent role in increasing CaOx supersaturation which finally predicts stone formation (Prohaska M et al., J Urol 199: 1262-1266, 2018). It has convincingly been demonstrated many years ago that CaOx supersaturation in human urines exponentially rises with increasing urinary Ox concentrations up to a concentration of 0.7 mmol/l, whereas CaOx supersaturation plateaus above urinary Ca concentrations of 5 mmol/l (Roberston WG et al, J Cryst Growth 53: 182-194, 1981). The excretion of oxalate, however, greatly depends on the amount of absorbed Ox which is influenced by the ingestion of calcium simultaneously with Ox-containing products. This was demonstrated by Von Unruh et al. (Ref. 56) as well as - under rather extreme conditions of Ox intake from natural foods - Hess et al. (Nephrol Dial Transplant 13: 2241-2247, 1998) in healthy volunteers. More recently, Sromicki & Hess (Urolithiasis 48: 425-433, 2020) have demonstrated that increasing the intake of calcium from natural foods/fluids simultaneously with any snacks/meals significantly reduced urinary oxalate and CaOx supersaturation in idiopathic calcium stone formers, although urinary calcium increased too. Similar findings highlighting the importance of timing of the intake of calcium (as supplements) were obtained by Domronkitchaiporn et al. (Ref. 60). These findings should be added to the review.
Thank you for these comments. We have revised the “Oxalate section” to address these comments and include the highlighted findings.
This includes a new first paragraph highlighting the relationship between oxalate and stone risk: “Hyperoxaluria is a risk factor for calcium oxalate stone formation by increasing urinary saturation of calcium oxalate, which has been shown to predict stone formation.[66] Whereas CaOx supersaturation plateaus above urinary calcium concentrations of 5 mmol/L, there is an exponential rise in CaOx supersaturation with increasing urinary oxalate concentrations up to 0.7 mmol/L.[67] Thus, oxalate intake is a significant dietary target for stone prevention.”
Additionally, the 4th paragraph of the “Oxalate” section has been significantly revised to underscore the relationship between the timing of dietary oxalate and calcium intake: “Although dietary oxalate is directly correlated with urinary oxalate excretion.[77], [78] intestinal oxalate absorption depends not only on dietary oxalate intake, but also on dietary calcium intake. Diets with higher calcium lead to reduced intestinal oxalate absorption, while restricted calcium diets are associated with enhanced oxalate absorption and increased urinary oxalate excretion.[56], [79] Indeed, Holmes and colleagues showed that dietary oxalate intake contributes up to 24-41.5% of urinary oxalate excretion, but on a high-oxalate, calcium-restricted diet, dietary oxalate intake can contribute up to 52.6% of urinary oxalate.[75] The timing of dietary calcium and oxalate intake plays an important role in determining the effect of dietary oxalate on stone risk. In idiopathic calcium oxalate stone formers, increasing dietary calcium at meal and snack times was found to significantly reduce urinary oxalate and CaOx supersaturations, albeit along with an increased urinary calcium.[80] This finding was validated in a 2-phase, randomized, crossover metabolic study among 32 healthy men who were given either 3 gm of calcium carbonate daily (all at bedtime away from a meal), or 1 gm of calcium carbonate with each meal.[60] Although urinary calcium increased in both groups compared to baseline, urinary oxalate significantly decreased when calcium carbonate was given with meals but remained unchanged when given at bedtime. Indeed, urinary saturation of calcium oxalate was unchanged when calcium was given with meals but increased with given at bedtime. These findings suggest that taking calcium with meals reduced urinary oxalate and also protected against the increase in urinary calcium seen with calcium supplementation.”
Excess vitamin C consumption and risk of kidney stones: the Swedish study of Thomas et al. (JAMA Intern Med 173: 386-388, 2013) should also be mentioned.
This study has been discussed in the second paragraph of the “Oxalate” section, where the relationship between oxalate and vitamin C is discussed: “In another study of Swedish men, vitamin C supplementation was further associated with a 2-fold increased risk of incident stone disease over 11 years of follow-up (multivariate RR1.92, 95% CI 1.33-2.77); however, the actual dosages consumed were not reported in this study.[69]”
Final paragraph: it should be mentioned that so far NO study ever has demonstrated a significant effect of oxalate restriction on stone recurrences, as oxalate is almost unavoidable. The clinical key point is to reduce gastrointestinal Ox absorption by simultaneous intake of calcium with meals and snacks, most likely together with the reduction of fructose-containing products in order to reduce urinary oxalate form metabolic origin.
These points have been added to the last paragraph of the “Oxalate” section: “Of note, no published study to date has demonstrated a significant benefit of dietary oxalate restriction on actual stone recurrence. However, co-ingestion of calcium-containing foods with oxalate-rich meals or snacks can prevent the rise in urinary oxalate and urinary calcium oxalate saturation, but with an increase in urinary calcium.”